# "The world is not a safe place": Representations of emotional distress, coping, and survival among young adults in South Africa

Sorcha Ní Chobhthaigh[1]*, Smanga Mkhwanazi[2], Samantha Willan[2], Jenevieve Mannell[1], Laura Washington[3], Andrew Gibbs[1,2,4,5], Rochelle A. Burgess[1,6]

1 Institute for Global Health, University College London, London, United Kingdom, 2 Gender and Health Research Unit, South African Medical Research Council, Pretoria, South Africa, 3 Project Empower, Durban, South Africa, 4 Department of Psychology, University of Exeter, Exeter, United Kingdom, 5 Centre for Rural Health, University of KwaZulu-Natal, Durban, South Africa, 6 Department for Social Work and Community Development, University of Johannesburg, Johannesburg, South Africa

* sorcha.nichobhthaigh.21@ucl.ac.uk

## Abstract

In contexts of extreme adversity and oppression, trauma exists as an open-ended ongoing threat, requiring us to recognise the ways people cope as adaptive survival mechanisms shaped by external hardship. Through a Black feminist lens, we explored the narratives of 17 young people in KwaZulu-Natal, South Africa, detailing their day-to-day experiences navigating hardship and potentially traumatic events, collected as part of a co-produced community-based initiative. Using grounded thematic network analysis, we offer an understanding of the young people's emotional experiences and the spectrum of strategies they use to cope. *Feeling through Survival consists of* five themes representing emotion experience of trauma, ranging from heaviness and fear to joy and pride. *Coping as an Act of Resistance* encompasses eight themes depicting young people's efforts and subtle agentic acts to persevere; including collective patterns, sacrifices, adaptive efforts to create a sense of safety, escapism, breaking free and boundary-setting, support systems, sense-making, and for some, meaning-making. Our analyses provide insight into the complex experience of trauma and survival, where multiple, at times, seemingly conflicting, realities co-exist. By adopting a non-judgemental approach to coping, we move beyond simplistic dichotomies of risk or resilience, deepening our understanding of what it means to live and cope amidst persistent adversity. Our findings highlight the importance of contextualising individuals' emotional and behavioural functioning within the circumstances that drive their efforts to cope. This understanding necessitates a shift in approaches to therapeutic interventions with individuals and communities facing open-ended trauma, particularly in resource-constrained settings. Recognising the risks of assuming safety or closure from trauma, interventions

**Data availability statement:** Availability of data used in the study would be subject to permission by the Health Research Ethics Committee and provincial authorities that approved the original study. This is a recently completed study and the dataset will initially be used for capacity development among the emerging researchers on the study team. Thereafter access to a de-identified dataset is available upon reasonable request. Requests should be sent to the convenor of the South African Medical Research Council's Research Ethics Office, Ms Adri Labuschagne (Adri. Labuschagne@mrc.ac.za), for consideration. Guidelines for applications and related materials are available at: https://www.samrc. ac.za/research/rio-research-ethics-office (Adri. Labuschagne@mrc.ac.za), for consideration. Guidelines for applications and related materials are available at: https://www.samrc.ac.za/ research/rio-research-ethics-office.

**Funding:** This work was supported by the UKRI-MRC Global Challenge Research Fund, Contexts in Health award (MR/T029803/1 to RAB, SM, SW, JM, LW, AG) and the UK Medical Research Council (MR/N013867/1 to SNC). The funders had no role in study design, data collection and analysis, decision to publish, or preparation of the manuscript.

**Competing interests:** The authors have declared that no competing interests exist.

should be re-oriented towards supporting navigation of ongoing traumatic contexts, addressing barriers to accessing safety, and facilitating community empowerment.

## 1. Introduction

With increasing global awareness of the risks to mental health and the burden of distress experienced by young people, there has been a growing prioritisation of research on interventions, that will eventually shape shifts in practice and service development. Understanding local and young person-led ways of communicating distress and coping is central to accurately identifying mental health difficulties and providing structurally and culturally-safe care. The reliance on diagnostic instruments and standardised measurement tools developed in high-income Global North countries, may not only miss locally manifested aspects of experience [1] but also risk perpetuating stigma and misunderstanding [2]. In turn, implementing context and culture-blind mental health interventions in low-resource settings also risks falling short of meeting the needs of communities, as well as exacerbating distrust in professionals, further deterring help-seeking [3,4]. However, understanding representations of distress and coping is not enough to fully understand and honour lived experience. Moving beyond 'Western'-bound or Eurocentric individualistic ideas of coping and resilience requires engaging with the multiple realities of navigating day-to-day life in the face of ongoing adversity. Crucially, this necessitates bridging the language gap between young people's framing and expression and the languages used by providers and researchers. In many cases, this will require that practitioners learn and engage with new 'languages' that enable them to meaningfully engage with the complexity of young people's lived experiences and responses to distress while resisting the urge to pathologise or ascribe clinical language to make sense of these experiences. In the interim, to support this process, there is a need to hold both perspectives, translating the voices of young people into language accessible to practitioners, while simultaneously creating space for the evolution of practice that embraces nuance and contextualisation.

Prioritising an understanding of the complexities - and at times, the conflicting nature - of experiences, rather than imposing external conceptualisations of challenges, aligns with critical development studies, which problematise westernised notions of agency [5,6]. As well, Black feminist literature witnesses how people navigate complex oppressive realities, without prescriptive value judgements about the forms or perceived legitimacy of those survival strategies. For instance, recent work acknowledges and values the agency asserted by young Black men in Caribbean countries who successfully navigate poverty and the afterlife of slavery through participation in 'scams' or pyramid schemes [7]. When recognising the lived realities of communities who have endured state-sanctioned violence and oppression, certain frameworks can replicate oppressive and othering lenses, contributing to further oppression. An overreliance on binaries in mental health, alongside conditions for accessing care (i.e., access to housing, or 'zero-tolerance' approaches) can

be particularly problematic in contexts where trauma has not ended, and even more so, where the trauma is ongoing or rooted in socio-political-historical landscapes and languages where care is situated [8]. In this paper, we seek to advance the body of literature around coping and survival linked to the trauma and mental distress of young people living through adversity, using a qualitative analysis of data collected as part of the Siyaphambili Youth Co-development project, in South Africa.

## 1.1. Trauma

Often there are implicit assumptions in our approaches to understand reactions to trauma, namely, that the precipitating event that triggered the reactions has ceased, and that individuals' reactions are indicators of internal suffering, rather than a function of ongoing external hardships. This has shaped extensive debates on the best ways to approach and define trauma.

From a bio-medical perspective, many, although not all, individuals with exposure to potentially traumatic events meet some diagnostic criteria for mental health diagnoses such as anxiety, depression, or post-traumatic stress disorder [9]. However, there is wide heterogeneity in individuals' responses to potentially traumatic events [10] and conceptualising complex trauma and the impact of exposure in diagnostic or medicalised terms may not always be meaningful or useful [11]. Black feminists and scholars, such as Sizani Ngubane [12], and Jessica Horn [13] assert the necessity of rejecting the boundaries created by the use of limiting diagnostic categories that are not centered within the worldviews of the people to whom they are applied. As has been argued by writers within cultural and critical psychological domains it would be more useful and meaningful to seek to understand local, contextualised, expressions of distress and efforts to cope in order to inform interventions that bolster these efforts, particularly in low resource settings. As we navigate bridging the gap between understanding and communication we shift away from rigid categories while recognising the validity in labelling potential risks and impact. Through the lens of trauma, we intend to facilitate understanding, in the absence of which risks invalidating, unsupportive responses and potentially ineffective or even *unsafe* interventions.

Broadly, trauma refers to the internal experience, the emotional and physiological response, to an overwhelming experience, adversity, or absence of vital support – that is, it is not defined by an event itself but by the impact of the event and the imprint left on the mind, brain and body [14,15]. Complex trauma extends this approach to encompass chronic exposure to (potentially) traumatic events or adverse experiences, which may include repeat experiences of or prolonged exposure to abuse, neglect, interpersonal violence, community violence, racism, discrimination, as well as systemic poverty and war [16,17]. The concept of continuous traumatic stress, coined in the context of apartheid South Africa [18], further differentiates the contexts of ongoing traumatic stress conditions from historical trauma that has since been escaped [19].

In cases of open-ended trauma, the focus of this paper, ongoing exposure and perpetual threat of future exposure is a defining feature, where exposure to the trauma-inducing factors remains indefinite. As such, reactions to enduring potentially traumatising environmental realities in the present need to be understood not as symptoms, but as adaptive and normative responses to extreme conditions [20]. This drives us to meaningfully situate threats as social factors that individuals are consumed by surviving and to acknowledge how socio-political realities are deeply intertwined with and act as drivers of individual efforts to cope.

## 1.2. Trauma in the South African context

In South Africa, contemporary structural inequities and governance failures persist in the afterlife of colonialism and apartheid. While the formal apartheid system began to be dismantled with the unbanning of the ANC in 1990 and the transition to democratic governance in 1994, its effects continue to shape every aspect of society. These legacies are evident in the ongoing racialised economic disparities, spatial segregation, and deeply entrenched social inequities that persist despite

the end of formal apartheid policies. The continuous cycle of traumatic stress conditions are long-standing risk factors sabotaging economic, social, interpersonal, physical, and mental well-being [21,22] that perpetuates extreme levels of violence [23]. Violence between family members or intimate partners, within communities, as well as in collective response to oppressive environments, is consistently ranked among the highest in the world [24,25]. While nationally it has been estimated that 35.5% of women reported experiencing physical and/or sexual violence during their lifetime [26], for women living in informal settlements this increases to two-thirds in the past year [27], and activists assert rates of gender-based violence and femicide worsened across the country during the covid-19 pandemic [28]. Further, with youth unemployment rates of approximately 66.5% across South Africa [29] and the interplay of socioeconomic marginalisation with accessing quality education, real and imagined opportunities to break cycles of poverty established by colonialism are stifled.

In the province of KwaZulu-Natal, risks of exposure to extreme adversity and potentially traumatic events among young adults are amplified. Youth unemployment is over 50%, and high rates of violence sit alongside the impacts of climate events, such as extreme droughts, heat waves and floods. Ripple effects increase other health risks, impeding access to basic supplies and heightened food insecurity [30]. KwaZulu-Natal also has the highest HIV burden in South Africa, with a prevalence of 21.8% [31]. For those living in informal settlements the likelihood of experiencing a traumatic event is increased [32], as well as the ripple effects of normalisation of violence. Each of these individually impactful events often interact, resulting in a complex, synergistic impact greater than the summed effects [33]. Moreover, intergenerational transmission of trauma manifests through socio-economic, biological, relational, social, psychological pathways [34]. Impossible to disentangle, these contextual factors represent unavoidable risk of repeated exposure to synergistic adversities for young adults in this region.

## 1.3. Coping as routes to survival

Although coping refers broadly to the ways in which individuals survive challenging situations and navigate towards well-being, literature on coping has for the most part explored 'positive' coping strategies. That is, strategies that are deemed appropriate and acceptable to a group, profession, community, or wider society, such as seeking social support or engaging in activities. Explorations of the potential utility in coping strategies labelled 'negative' or 'unhealthy' by a bio-medical lens are typically framed as either symptomology or 'adverse coping' strategies. However, whether a coping strategy is 'adaptive' or 'maladaptive' is context-specific and determined by the wide variation of factors driving adversity. That is, over time and through repeated exposure, individuals develop internal narratives, emotional responses and patterns of behaviour that are *adaptive* to their environment. An inadequate attending to context may result in labelling coping as maladaptive or dysfunctional, pathologising adaptive and functional responses to adversity. In resource-limited settings, where access to safety, resources or specialist services may be constrained, there is also a potential risk associated with taking steps to address trauma, in ways typically deemed 'adaptive'. For example, disclosure of trauma without an appropriate response can lead to re-traumatisation, or rejection by peers, family, or community members who have not processed, or are not in a position to recognise their own pain [35], where collective avoidance has enabled them to persevere, and where discussions of the pain and trauma evoke feelings of shame. Moreover, whether a coping strategy is categorised as 'healthy' or 'unhealthy' is based on a perception in comparison to a deemed 'normal' [36]. As such, emphasis on 'positive' or 'healthy' coping strategies and not 'coping', not only fails to accurately capture the lived experience of individuals, it also risks pathologising 'normal' responses to 'abnormal' circumstances.

## 1.4. Rationale

In this paper, we explore how young adults (age 18–24) negotiate emotional trauma and survival while navigating ongoing social and structural risk factors for mental health difficulties as well as repeated exposure to potentially traumatic experiences in the everyday. By shifting focus to coping and survival, this takes into account the full spectrum of strategies

that allow individuals to 'cope' with adversity, without comparison to a deemed 'normal' or 'healthy' alternative. Shaped by Black feminist thought, we sought to depict young adults' representations of distress, coping and survival in the absence of mental health support in a community-based sample in rural and urban KwaZulu-Natal, South Africa.

## 2. Methodology

### 2.1. Ethics statement

Ethical approval was provided from two institutions, the South African Medical Research Council (EC041-10/2020) and University College London (9663.003). Youth Peer Research Associates (YPRAs) gave written informed consent to participate in the study before taking part. Safety planning, supporting service navigation and connections with community-based care were embedded into the project. All interviewers were trained in supporting YPRAs with strong emotions. If needed, interviews were paused or stopped, and the Project Empower social workers provided additional support or facilitated further support pathways.

### 2.2. Conceptual framework: Understanding coping and struggle through a Black feminist lens

Dominant approaches for engaging with young people's mental health in contexts of adversity often make sense of their needs and potential responses using conceptual frames and diagnostic tools grounded in the bio-medical model, rooted in 'Western' Eurocentric thought. To create space for a broader conceptualisation of emotional distress and coping, we approached this analysis through the lens of survival, in line with Black feminist praxis [37], framed by three processes: Open-Ended Trauma, Agency and Non-judgmental approaches to understanding coping.

Black feminist scholarship, in particular intersectionality, draws our attention to the necessity for counter-hegemonic knowledge production, which is focused on the transformation of people's lives, and a more radical politics of social justice which must be embedded in our critiques of mainstream approaches [38,39]. Considering mental health, this takes the shape of questioning the value of treatment and diagnostic paradigms in real-world contexts of people seeking to ensure their survival, particularly when living in bodies who experience multiple forms of violences. For example, diagnostic criteria for Post-Traumatic Stress Disorder (PTSD), a typically assessed condition in studies of young people's mental health within the DSM-V depends on the triggering event as being 'past-tense' [40]. However, as noted above, these framings are limited in their assumptions about the temporality of trauma. In many instances, long-wave social crises such as inter-generational poverty, the enduring impacts of colonialism and, in the case of South Africa, Apartheid, are past, present and future legacy, not 'post'. These active processes defy the notion of 'post-trauma' instead reflecting complex open-ended trauma that continue to shape the mental health and wellbeing of millions of Black people globally [41,42]. The open-ended nature of exposure makes diagnostic criteria centered on historical or once-off events ill-suited to the needs of young South African communities.

Agency, within much public health and mental health literature, is often focused on the ability for actors to take action, and the ability to feel as though one is in control of one's actions and choices in treatment [43,44]. Often though, these perspectives are anchored to westernised, and often decontextualised perspectives on what counts as meaningful 'action'. Instead, we orient ourselves to perspectives of agency rooted in critical feminist praxis, which prioritises the relational and the micro-acts of agency that are embodied in everyday survival. For example, early work of Sumi Madhok [45] argues marginalised women's agency is often overlooked by over-emphasising the ability to 'act freely' – when we should instead, be attending to women's decisions *not to act*, or small actions geared towards survival, as more meaningful examples of agentic action. Burgess and Campbell [6], as well as Willan et al. [46], argued that women's agency should always be considered in the contexts of their immediate realities, taking seriously the relevance and validity of the options available, the contexts and internal narratives underpinning their choice, or refusal to choose. Such perspectives align with concept of ambiguous agency which questions the kinds of agency deemed 'appropriate' for children and youth, and

the need to challenge the moralising discourses that simplify agency into isolated 'empowered' acts [47]. Combined, these perspectives on agency direct us towards the need to stay with the relational dimensions of survival; attending to how young people relate to and through social, political, economic, and familial contexts as they promote their survival [48].

Drawing from Mad studies, which includes the study of service user rejection of mainstream services and care as the gold standard, in favour of non-medical and social routes for care and support [49], we consider the ways in which people engage in behaviours that to one system may seem problematic, but within the context of an individual's reality - may be a necessary act in survival. We question the utility of labelling coping mechanisms when seeking to understand how individuals or communities *survive* within extreme socio-political circumstances. For communities navigating the realities of oppression, conflict and extreme inequity, reactions need to be understood as *protective* survival instincts. Ultimately, agency within conditions of continuous traumatic stress is the act of choice that enables an individual or community to cope and persevere.

### 2.3. Setting: Developing interventions *with* young people

This analysis draws on data collected as part of the Siyaphambili Youth ('Youth Moving Forward') project [50], a 4-year community-based initiative in KwaZulu-Natal Province, South Africa. In a partnership between the South African Medical Research Council (SAMRC), a South Africa non-governmental organisation (NGO) called Project Empower, University College London (UCL), University of Exeter and a team of 17 Youth Peer Research Associates (YPRAs), over 36 months, the project sought to co-produce an intervention to empower young adults and address the social and contextual factors that lead to violence and poor health. The YPRAs, aged 18–29 years (9 female, 8 male), were recruited between 17 November 2020–22 January 2021 from rural communities and urban informal settlements previously engaged with the research team to co-develop interventions that support young people's agency to modify HIV-acquisition risk. These young adults lived in communities faced with numerous challenges ranging from poverty, high levels of community unemployment, limited social support, violence, alcohol use, and poor mental health.

### 2.4. Data collection: Collaborative work with Youth Peer Research Associates (YPRAs)

Data for the current analysis was collected as part of the first and second phases of intervention co-development process. Phase One involved a range of activities to encourage the YPRAs to reflect on their lives, this included a modified photo-elicitation process and semi-structured interviews about selected photographs. This aimed to understand the social contexts that create overlapping risks for HIV, IPV and poor mental health as well as individuals' sense of choice and agency in the face of such risks. In Phase Two, YPRAs drew on personal experiences to develop fictional character stories (character creation) depicting daily life to facilitate sharing and exploration of issues related to agency and risk. They described their experiences of overlapping challenges, such as unemployment, drug and alcohol use, violence, mental health challenges, and structural exclusion and how these challenges have impacted on their lives and modes of survival. All interviews were completed in isiZulu by members of the wider Project Empower team - isiZulu speaking Black women and men from various backgrounds - who had been supporting and working with the YPRAs throughout the project, and were supported by LW. Interviews were translated into English prior to analyses. None of the YPRAs were in education or formal employment at the time (see Table 1 – pseudonyms used).

### 2.5. Analysis

Interviews and character creation exercises were analysed using Burgess' [51] *grounded thematic network analysis*, drawing on Attride-Stirling's [52] *thematic network analysis* with Mano's [53] addition of supra-global themes. The step-wise progression of basic themes into higher-order organising themes, maintains the connection to context and grounding in participant experience. The global themes encapsulate the shared threads cutting across experiences and contextual

**Table 1. YPRAs demographic information.**

| Name[*] | Gender | Age | Education completed[^] | Children |
|---|---|---|---|---|
| *Urban* | | | | |
| Masiduli | Female | 23 years | Matric | One |
| Nomasonto | Female | 22 years | Grade 8 | One |
| Nonhlanhla | Female | 24 years | Matric | None |
| Nonsindiso | Female | 23 years | Grade 11 | One |
| Thembeka | Female | 22 years | Matric | One |
| Nhlakanipho | Male | 29 years | Matric | Three |
| Ntuthuko | Male | 23 years | Matric | Two |
| Velani | Male | 22 years | *Not disclosed* | One |
| Kwanele | Male | 24 years | Matric | Two |
| *Rural* | | | | |
| Sthokoza | Female | 22 years | Matric | None |
| Nokuthobeka | Female | 20 years | Matric | None |
| Nomthandazo | Female | 20 years | Matric | None |
| Philisiwe | Female | 22 years | Matric | One |
| Nkanyiso | Male | 22 years | Matric | None |
| Mandla | Male | 20 years | Grade 11 | None |
| Nkosinathi | Male | 24 years | Matric | None |
| Sandile | Male | 24 years | Matric | None |

[*]Pseudonyms.

[^]Matric is final year of secondary education.

specifics, the addition of the supra-global themes, allows for clustering of global themes without the loss of complexity at the higher levels, lending a more nuanced understanding and holistic interpretation.

Following familiarisation with the data, codes were identified from the transcripts. Codes were then grouped into basic themes. Basic themes were then grouped into wider organising themes, which were grouped into overarching global themes (see S1 Table). Given the focus on local representations of distress, resilience, and efforts to cope, contextual information was retained and reflected upon at every stage. This is also critical for a piece of work embedded within a larger program of co-produced research. These efforts seek to limit the positioning of researcher positionality and perspective as truth over and above the narratives generated by everyday actors engaging in programs of survival. We are also keen to avoid the ways in which psychiatric paradigms have been critiqued for silencing of lived experiences, and as such our analytical approach seeks to incorporate the complexity of everyday life within the process of theme generation to maintain a closeness to the data as presented by YPRAs. These themes represent the range of and diversity in experiences among the YPRAs, rather than disaggregating for a particular "type" of individual. In line with these interests, we did not complete a gendered analysis, but do reflect on any differences we noted between participants throughout.

## 2.6. Reflexivity

Following on from the above, we were constantly aware that the data we were analysing was the product of translation and transcription into English. The first author (SNC), a white Irish former mental health clinician, working across contexts and countries, joined the team after data collection, as such, she exists outside of the local context this paper explores. Mindful of the importance of the analysis remaining contextually rooted, two lead authors (SNC+RAB) engaged in a process of close dialogue throughout code and theme development, consulting and integrating feedback from the wider

research team in order to prevent over- or mis-interpretation and ensure the findings felt accurate and meaningful for those closest to the context. Further, mindful of the dynamics and range of perspectives on the team which influenced our analysis, we include a note on the positionality of the research team. RAB is a Black feminist woman who has studied and worked in the South African context across the last decade. JM is a sixth-generation Canadian of Irish, English, and French heritage, descended from early white settler communities. While not South African, she comes to the paper with a post-colonial feminist background and several years (since 2009) of in-depth work in the South African context. AG is a white man who lived and worked in South Africa from 2006-2022 and continues to undertake research focused on young people's lives in these contexts. SW is a South African feminist, who has always lived and worked in KwaZulu-Natal and the social justice sector, nonetheless as a white woman she carries certain privilege which must be continually checked. SM is a Black man in his mid-30s from South Africa. He works with young people and has a keen interest in studying the experiences of young men through qualitative research. However, he recognises the importance of regularly examining his own privileges, as he comes from a different social class. LW is a white South African woman who has worked in KwaZulu-Natal to understand the impact of historical and contemporary oppression on the lived realities of women and men.

## 3. Findings

The YPRAs' reflections on their lived experience and accounts of daily life were captured by two supra-global themes. *Feeling through Survival*, incorporates multiple global themes offering insight into specific emotional dimensions of coping. *Coping as an Act of Resistance,* incorporates the spectrum of strategies used by young people in responding to distress. Supra-global themes were informed by 13 global themes. Overall, the narrative of the analysis allows us to describe the contextually driven nature of coping behaviours engaged in by young people. Each of these global themes are described in turn below, anchored to quotes to reflect contextualised accounts of what it means to survive adversity and navigate ongoing emotional trauma. Tables summarising our thematic networks can be found in Supporting information.

### 3.1. Feeling through survival

This supra-global theme provides insight into the emotional landscape of young people as they etched out survival in their everyday lives, combining five global themes that give space to the multiple and, at times, conflicting truths borne of sitting with adversity and trauma.

The first global theme: *"It was fine... but not totally" - the weight of trauma* provides a rich descriptive narrative into the contexts of emotional lives and sense of pain, heaviness and isolation as well as the internalisation and somatisation of pain that frequently comes along with potentially traumatic life events. Shared across genders and contexts, YPRAs frequently described feeling disempowered, pained and alone, which appeared associated with experiences of trauma at individual, interpersonal and community levels. A palpable sense of deflation within the wider societal context was evident, particularly in relation to gender-based violence, for example, when describing her mothers' response to repeated dismissals by her in-laws when she tried to report her husband's abuse towards her "*she kept quiet because there is nothing she could do*"(*Nonsindiso-urban-female-23years*). YPRAs also described feeling dejected around inequalities in access to emergency services, unemployment rates, community violence, access to education or classism within communities, as one YPRA shared "*they* [the community] *didn't take it seriously as you know that people treat things with priority if it happens to families with high profiles who have their own money. So they didn't take this one seriously because she got stabbed at her house* [neighbour killed by her husband]*, it took a lot of time to get a car that will take her to the clinic...So this shows that since I also have nothing if something were to happen to me I will be like her.*" (*Philisiwe-rural-female-22years*).

YPRAs reflected on instances of self-blame and self-doubt, disclosing critical internal narratives and a burden of feeling 'not good enough'. The global theme *"I thought that maybe I was wrong too" - the internal spiral*, provides a picture of the internal processes and rationalisation used by young people in accounting for their experiences. In some cases, this

was linked to reflections on external ideals as well as wider societal expectations, reinforcing a perceived loss of dignity, or at worst, worthlessness. These internal processes aligned with cognitive thinking patterns such as rumination, mental replay, and escaping through imagination, for example, "*if so and so was around things would be different*" (*Velani-urban-male-23years*) or "*I would wish my father, or my uncles, were still alive, maybe I would have lived with them.*" (*Nomasonto-urban-female-22years*).

The third global theme, *"It's all too much" - Fears and expectations in everyday life,* reflects YPRAs' experiences of anxiety and overwhelm as they cope with feeling unsafe in their environment and the weight of scrutiny and expectations from their community. Male and female YPRAs across urban and rural contexts described threats to their physical safety in their day-to-day life, including everyday violence, but particularly among women detailing interactions with men when in the community. As noted by one YPRA: *"It's safe inside but what affects us is that we don't have street lights, it's dark if you walk from home alone. Obviously, there are always boys by the gate, they can even pick pocket you if you're walking alone because there are (no) street lights so we are not safe as women..." (Thembeka-urban-female-22years).* Additionally, they described unsafe and unstable working conditions, threats to their mental well-being, such as anticipated social judgement or rejection, for example, one YPRA explained *"Looking at other people's relationship will make* [it harder] *and having nothing to talk about when it comes to my relationship status is very hard on its own." Interviewer: "So, you prefer having something to talk about to your friends even though you are not happy in your relationship, just for other people to know you in a relationship" YPRA: "Yes, because sometimes as girls we look down on each other. People always talk 'cause you will find that if a relationship don't last then they will think you have a problem because of a relationship not lasting" (Masiduli-urban-female-23years).* Experiences of overwhelm were shared across genders - "*I was studying but it was hard because the workload was too much and the baby was here, the person who was looking after the child only came during the day and not at night. I had to look after the baby when it's during the night, so it was hard to study. The child would be crying, so my grade 12 was ugly.*" *(Philisiwe-rural-female-22years)* but for men in particular, a sense of feeling weighed down by pressures and expectations was evident: *Interviewer: "So, there's pressure to a young man on what he eat or buy in house when there's a woman?" YPRA: "Too much pressure." Interviewer: "Do we put this pressure to ourselves as young men or is it women who put this pressure on us?" YPRA: "It's them." (Nhlakanipho-urban-male-29years).*

Shared across genders and contexts, YPRAs also described pervasive distrust of others, excessive vigilance and a never-wavering readiness to defend themselves as they navigate day-to-day life. This was illustrated by the global theme, *"Eventually… people show their true colours" - Prepared for the worst: Bracing for betrayal & rejection*. The sensitivity to threat, in reaction to perceived criticism, disrespect, rejection, or even unfamiliar situations, in turn, led to and reinforced efforts to self-protect, overtly for example, *"They had both taken a* [shared] *taxi from* [the township] *and the girl was drunk and started a topic in the taxi about ''boys are dogs'' and then my brother argued that he's not a dog so the argument escalated and he jumped off* [got off the taxi] *where the girl was going to jump off* [get off] *in front of her house,* [and that] *is where he stabbed her" (Masiduli-urban-female-23years)* - or covertly, by keeping others at a distance, keeping secrets and believing that it is safer to assume the worst;*" - "Even my best friend doesn't know what I'm doing... 'cause I don't even trust him 'cause when we are talking I can tell that, no, if they were to know, I'd end up being a debate when I'm not around. There are things that you have to avoid from happening 'cause they will hurt you..." (Velani-urban-male-22years).* This threat-protect cycle also seemed to provide insight into YPRAs' relational narratives, as many shared a deep sense of distrust and guardedness as well as the understanding of love and pain as intertwined.

Nevertheless, alongside these depictions of pain and adversity, all YPRAs shared instances of joy, laughter and gratitude as well as reflections on the importance of feeling respected and valued by others. The global theme *"We all have a story to tell growing up, I'm just grateful we found each other in the end" - Appreciation for life & Sense of self-worth,* encapsulates the sentiments of hope for the future and appreciation for the little things that make life worth living, as one YPRA reflected *"Even though it was not the same being raised by my gran instead of my parents because she was old. But she made sure to take care of me always; and I learnt to take care of her. I did not feel the gap that my mother left*

*because my gran was there, and she was everything to me.” (Nomthandazo-rural-female-20years).* YPRAs highlighted the importance of celebrating and sharing successes, such as passing school exams, getting paid, being a parent, reconnecting with family, having someone pay *lobola* (“bride price”), winning sports, or even instances of feeling respected by others, whether it be kids, peers or community members, for example,*“It was at home, it was my first day, I got paid. I was thankful. I got the job, so I was thankful….it was a ritual…I was asking* [the] *ancestors* [for the things that] *what I want…. We do rituals... Sometimes you have to say things to your ancestors and thank them. I was happy that my family was together.” (Nhlakanipho-urban-male-29years).*

### 3.2. Coping as an Act of Resistance

*Coping as an Act of Resistance* is made up of eight global themes depicting YPRAs' journey of survival, the realities of coping with pain and trauma, and in some cases, simultaneously, experiencing growth and healing. This supra-global theme encompasses the full spectrum of efforts made by young people to navigate ongoing adversity and risk, recognising that any effort to persevere, to self-protect, to *cope,* is in itself an act of resistance against structural oppression and social injustices.

YPRAs described repeating patterns of behaviour, individually and collectively, in an effort to survive and meet their foundational human needs as captured in the global theme *“You are never mugged by someone who you do not know, it's always someone you know” - Trapped in the cycle together*. That is, individuals *cope* with income, food and housing insecurity through a complex web of actions and reactions, ranging from direct involvement in unconventional or illegal methods to obtain resources, to passive acceptance of such activities within their community. These cycles of shared experience are normalised, manifesting in various ways, whether through active involvement, interpersonal tactics to assert power or feel a sense of control, or passive bystanding as part of unspoken agreements within the community. YPRAs frequent descriptions of reluctance to intervene suggest an unintentional complicity in sustaining these cycles, born out of self-preservation, for example *“So, there is nothing we do, we just keep quiet. You get mugged and we just keep quiet. Even when someone get mugged in front of you, you do not say anything but you keep quiet.” (Thembeka-urban-female-22years)* and *“we don't get involved. I do not stop any fight there, I just watch.”(Velani-urban-male-22years)*. At the same time, some YPRAs shared sentiments of ‘you give what you get’, indicating a sense of entrapment within the cycle and a protective strategy – as described by this interaction *YPRA: “I'm not a punching bag therefore I will fight back” Interviewer: So in this circumstance did you also hit back?” YPRA: “He slapped me, and I did the same,” (Nomasonto-urban-female-22years)*. Mirroring a collective ‘fight’ or ‘flight’, this may be mis-perceived as acceptance or fueling of an environment where the alternative may be perceived as risky, futile or more dangerous. Ultimately, the complexities and pervasiveness of precarity shape the ways in which young people assert power or seek survival.

Similarly, the global theme, *“Sometimes you need to sacrifice in order for you to survive” - What it means to Survive* attempts to capture the determination to keep pushing forward despite extreme hardship and potential feelings of despair. The young adults shared narratives around taking things into their own hands and the realities of making trade-offs in meeting their needs, such as hustling to earn money to buy food at a cost of leaving education early - *“...here I am today, how I survived I also ask myself but hey I did survive in different ways. Sometimes you need to sacrifice in order for you to survive, it doesn't matter what people say. As long as you don't do bad things and rob people, that's good. I did a lot of things.” (Velani-urban- male-22years)*. They also described attempts to solve potentially emotionally activating and messy problems with straightforward logic-oriented solutions, for women in particular, viewing dating as a form of income, or finding a second partner as a way of resolving issues with the first. They shared examples of ‘playing along with’ men to manage risk *“We feel oppressed, one para (thug) always called me his girlfriend. He robs people and also beat them, if he says “hello, my girlfriend” and you do not respond he will slap you. I would always say “yes” to that. If he would say “here comes my girlfriend”, I would smile because if I would be grumpy or didn't say anything I would be slapped, or he would rob me. For me to be safe, I would say agree when he says I was his girlfriend so that he wouldn't rob me.”*

*(Thembeka-urban-female-22years).* This prioritisation or compartmentalisation of needs enables coping with the most immediate or potentially threatening issue.

The global theme *"You could think and say they are abusing you, but they are building you"* - *Establishing safety in an unsafe world* reflects YPRAs' efforts to adapt to their environment, to manufacture a sense of safety and stability. This recognises the realities that in some circumstances, resignation or avoidance may be perceived as the safest options. YPRAs displayed efforts to modulate potential feelings of distress, by dismissing or down-playing experiences, glossing over or appearing to brush past discomfort, creating an internal narrative that maintains stability. While both men and women employed strategies of deflection, 'playing along', or minimisation, women also described more overt denial and rationalisation. These acts represent adaptive techniques to manage risk, whether physical, psychological or to prevent triggering a complex or escalating situation or mental health crisis. For example, when asked about potentially difficult experiences YPRAs reply, "*I won't say it's bad because it was sorted*" (*Ntuthuko-urban-male-23years*), "*Ahhh I am used to it, you get used to it, I do not have a problem*" (*Thembeka-urban-female-22years*) or "*It was really painful but it was beyond our control, I got closure…*" (*Nonsindiso-urban-female-23years*).

Across genders and contexts, YPRAs also described overt escapism, tactics to disconnect from their realities and attempts to 'shut-off' pain, particularly through substance use - "*when people are high on them* [drugs]*, the person forget what they were doing the previous day…something like that. That's why they say they are for depression*" (*Kwanele-urban-male-24years).* They described overt acts of emotion and intimacy avoidance, by numbing or blocking opportunities for vulnerability or closeness, as stated by one YPRA "*This thing of being around people leads to gossiping about other people and then you end up fighting, I am avoiding things like that. I am always alone...*" (*Philisiwe-rural-female-22years*). These efforts to feel a sense of relief and protection from the difficult emotions and unsafe circumstances were captured in the global theme *"All my problems… just vanish"* - *Distancing as a means of self-protection*. Escapism and avoidance act as necessary forms of protection against the continuous threat of adversity and simultaneously create a sense of freedom which may refuel perseverance efforts.

YPRAs consistently described the importance of asserting a sense of agency in their own lives, feeling empowered and working towards a future of their choosing. In the global theme, *"You have to put a stop to it"* - *Breaking the Cycle & Becoming a leader in your own life,* they spoke to rejecting expectations, establishing a sense of self beyond the limits that were imposed on them. Although this appeared across contexts, for women this was particularly salient when reflecting on relationships, from setting boundaries and limits in romantic relationships, choosing to parent differently to how they had been raised. For example, one YPRA disclosed "*I had to make a decision and to put an end to the abuse. I tried to end the relationship a couple of times but I got cold feet because I was scared of his reaction. So, in the end I had the courage to end things with him for good and he decided to also abandon our child as well. So, I think all stakes rely with* [the burden falls on] *the person who is abused and put a stop to it." (Nonsindiso-urban-female-23years).* While for men breaking the cycle manifested more in accepting responsibility for themselves and a willingness to doing things differently - "*hmmmm….I just decided and it was not like something happened …no, I just decided that I have to be straight and focus*"(*Nkosinathi-rural-male-24years*). This represents a more active, rather than passive stance, suggesting that in certain moments or circumstances, when feeling more emboldened or have the capacity and access to resources, YPRAs are enabled to act in line with their goals, with greater authenticity. This moves beyond mere survival, where the circumstances allow them to assert agency in a more overt or external way.

Moving beyond survival, circles of support, guidance and feeling a sense of belonging were central to the young adults' descriptions of navigating day-to-day life, encompassed by the global theme *"We help each other… We understand each other"* - *Support systems & Collective care*. YPRAs reflected on the role of support systems, seeing the humanness in each other and ultimately knowing you're not alone - "*Sometimes you think too much and even have suicidal thoughts, things like that…whereas talking to people helps as they tell you how to handle each situation*" (*Sthokoza-rural-female-22years).* While both men and women spoke about the benefits of guidance and knowledge sharing, particularly from

older members of the community or from those perceived as more successful, role models appeared to hold particular significance for men. Moments of connection, caring and holding space for each other's pain alluded to the human need to be cared for, and particularly for women to feel accepted and understood. Not only does social support prevent loneliness or alienation, it also offers opportunity to build resources, potentially fostering other ways of coping.

Across genders and contexts, YPRAs also described potentially healing practices, creating life narratives, making sense of their experiences, building awareness of their emotions and shifting from reacting to responding. The global theme *"You heal when you cry" - Power & Strength in Vulnerability,* captures the vulnerability and conscious effort required to work towards healing, seen in the young adults' growing awareness, understanding and empathy for themselves and others, as well as the ability to express, make sense of and regulate their emotional reactions. One YPRA shared *"I met new people and learned a new life. I learned things I didn't know, so when I was coming across people, I saw that everyone has problems. So, we all have problems, it's different how we solve them." (Mandla-rural-male-20years).* This sense-making, taking steps to build self-awareness, understanding of their experiences and emotions, learning how to regulate their emotions and increasing empathy enables individuals to process experiences, manage reactions and choose alternative coping responses in the short-term. Over the long-term, these practices also help establish new routines of healing as individuals work through cycles of grief and shed previous strategies – for example, *"That is my alone time and I need to enjoy it and have a peace of mind and eat what I want. If I feel like crying, I do." (Philisiwe-rural-female-22years).*

Some YPRAs even mentioned moving beyond sense-making, where the focus is on gaining insight into internal narratives and emotional attunement, to meaning-making, and connecting with a sense of purpose and connection, including aspects of parenthood or deepening relationships with religion and spirituality. These processes were encompassed in the global theme *"You don't have to be rich to be able to play a role to your child" - Purpose & Meaning-Making*. Both men and women described acceptance of their realities and letting go of things not in their control, for example *"If I would get dumped, I would accept because you can't force someone to love you if they don't."(Mandla-rural-male-20years)* while developing new perspectives on their life goals and values, *"the reason I was scared to have a child was the child maintenance. I stay at home and if I have a baby which will need everything, while I have nothing. I just think the baby formula is expensive and also the diapers, all of that and also the baby's clothes. But looking at how I am now, I can have a baby, I can be able to contribute with the little that I have."(Nkosinathi-rural-male-24years).* Having meaningful outlets and things to do also seemed to provide a source of hope and connection for the young adults who had access to them, *"Another way I have fun is going to church... I am always happy when I am at church. I also go out sometimes but I no longer drink when I'm out, I just go for the sake of my friends" (Nokuthobeka-rural-female-20years).* Coping by way of practicing acceptance, or channelling energy into sources of meaning symbolises an element of freedom from constraints that dictate other methods.

## 4. Discussion

### 4.1. Summary of findings

Our work illuminates the experiences of young adults in South Africa navigating longstanding adverse socio-political-economic circumstances. Survival narratives offered insight into two key aspects of YPRAs experiences: the internal emotional experience triggered while navigating unrelenting insecurity and instability, and attempts to cope with the realities of repeating cycles of trauma. These emotional experiences included not only pain and feelings indicative of anxiety but also, the ability to find pockets of lightness, joy and pride. Alongside this, the accounts of day-to-day life offered an opportunity to gain a deeper understanding of the diverse spectrum of coping strategies that allowed the YPRAs to persevere in the face of complex and enduring socio-political-economic-historical realities, ranging from those typically characterised as 'positive' to those often judged as 'negative'. While we observed some subtle gender differences, overall, in our sample, YPRAs described shared experiences and coping strategies across genders.

Young people describe pain, feelings of isolation and disempowerment associated with internalised, interpersonal and structural factors. While many young people described symptoms typically associated with PTSD, such as hyperarousal, sensitivity to threat and feeling overwhelmed, their experiences differ in that the trauma remains ongoing, making it impossible to enter the 'post-traumatic' phase associated with PTSD diagnoses. Instead, they remain in a continuous cycle of complex trauma. At the same time, there were instances of sharing successes as young people reflected on the sense of self-worth they felt when being heard, seen and respected by others and expressed gratitude for family, friends and faith. These experiences echo Post-Traumatic Growth, which refers to *positive* changes experienced by individuals after enduring trauma, such as greater appreciation for life, relationships, sense of personal strength or spirituality, that co-exist with distress [54,55]. However, centrally, there is no "post-traumatic" when the threat of trauma persists. In contexts of ongoing violence, oppression and instability, interventions must be adapted to acknowledge the open-ended, continuous, nature of the trauma. This requires shifting focus to navigating the present, recognising that engagement, pacing and expectations for 'progress' must be adjusted accordingly, meeting the individual where they are at, while allowing space for processing the past if and when makes sense for them. In other words, therapeutic interventions may take the form of 'containing the pain' rather than sole focus on 'healing the wounds', acknowledging the limits imposed by navigating continuous adversity.

Descriptions of day-to-day life offered a raw reflection of the emotional, behavioural and practical efforts, the agentic acts that enabled young people to persevere within complex social circumstances. Among these, they described coping strategies that provided short-term relief or escape, but which also had simultaneous harmful ripple effects on personal health, interpersonal relationships and wider community well-being. For example, this included substance use which while potentially harmful, can act as a coping strategy for escaping distress or facilitating social connection. Substance use as coping has been well documented among survivors of intimate partner violence [56], lifetime trauma exposure [57] and oppression [58]. These 'harmful' practices often serve critical functions and expecting individuals to stop using these strategies without ensuring that individuals have access to, or can realistically engage with, alternatives that meet the same needs can lead to additional harm. For example, no longer engaging in shared substance use may result in a loss of social support and connection, leaving the individual isolated [59]. As such, approaches need to be tailored to fill the gaps left when harmful practices are phased out, addressing not only the immediate needs of safety and security but also the emotional, social, and practical needs that may otherwise go unmet. Interventions that acknowledge the function of coping behaviours have been argued to promote empowerment and autonomy [60]. Substance use interventions, therefore, may need to embrace a harm reduction stance, recognising that while reduced use or abstinence might be a longer-term goal, emphasis should instead be on reducing harm as a more realistic and compassionate approach in the context of ongoing trauma.

Our findings highlight the dual burden of daily stressors, alongside the anticipation of further stressors, in situations where safety from potentially traumatic events cannot be ensured. For example, navigating employment insecurity in a socio-political-historical context with long-standing high rates of unemployment, can trap individuals and communities in a heightened emotional state as a function of being unable to establish food, housing and income security. This oppression-related vigilance, mirroring 'racism-related vigilance', that is, preparation for and anticipation of discrimination [61], offers insight into the complexity of coping responses that are adaptive -serving a survival function- and simultaneously amplifying stress [62]. Similarly, when potentially traumatic events become daily occurrences, coping strategies may include masking emotions, deflecting attention from uncomfortable topics, or adopting a posture of apathy as a means to survive or a fear-driven effort to not stand out. In such contexts, young people may become attuned to discern when it is safe to feel, safe to express, feasible to resist or safer to submit [63]. While adaptive in the moment, these strategies often perpetuate cycles of trauma and isolation. This lends support to arguments to recognise chronic oppression as trauma [64] and the subsequent compounding impact on mental health [65], with ripple effects on physical health, cognitive and executive functioning. Without careful consideration of oppressive contexts as trauma-inducing, we risk shaming individuals or communities for not always coping in 'palatable' ways, perpetuating cycles of violence and reinforcing fears

around disclosure. Interventions which embed the principles of trauma-informed practices [66,67] may reduce the risk of re-traumatisation and avoid recreating harmful power dynamics that mis-place blame on individuals navigating traumatic circumstances.

However, within resource-constrained settings further consideration needs to be given to challenges and opportunities for ensuring safety. Providers seeking to work in trauma-informed ways must navigate barriers to creating 'safer' settings, which may feel aspirational in contexts where trauma is structurally embedded, and therapists themselves may share these realities. Without safety and sufficient resources to manage potentially de-stabilising pain which may arise during therapeutic interventions, there is a risk of re-creating harmful power dynamics, perpetuating further violences, and shaming individuals and communities for not 'getting better'. That is not to say that mental health treatment should be withheld until after structural barriers are addressed; psychological interventions can still be beneficial [68,69]. Again, what matters is adapting interventions and expectations to contextual realities. Approaches that honour survival efforts and facilitate growth may reduce the risk of re-traumatisation generated by therapeutic practices when a foundation of safety is out of reach [70]. Embedding trauma-informed and harm reduction principles necessitates actively critiquing and addressing oppressive political and structural realities. Crucially, there is a need for parallel critical, context-specific interventions that simultaneously address the impacts of individual and community trauma, support economic, social, and political well-being of those individuals and communities while targeting the manifestations and root causes of oppression [65]. This tandem approach is necessary because trauma and oppression are intrinsically linked. As such, therapeutic approaches must be implemented alongside and in service of broader efforts for change, including livelihoods approaches [71], skills development [72,73], community resourcing and advocacy for meso- and macro-level changes [74]. These approaches should never be devoid of efforts to dismantle the oppressive structures driving mental ill-health and redistribute power, thus enabling access to good mental health.

Finally, it is essential to avoid falling into a lens of tragedy or pity, as this too imposes external judgement and undermines the joy, resilience and power individuals and communities demonstrate – the act of resistance in finding pleasure, peace and purpose amid oppressive circumstances. Recognising the everyday acts of agency, both passive and active choices of coping in extreme circumstances, and crediting the act of survival itself, supports the development of a sense of personal control and self-efficacy. Acknowledging that joy and meaning are acts of resistance reinforces perceived power. The iterative process of empowerment likely unfolds gradually through experience, interactions, and reflection, rather than an achievable 'state'. This aligns with the empowerment process model which highlights the importance of fostering a sense of power, through meaningful goal setting, knowledge, awareness and skill building, facilitating network building and drawing upon community supports [75]. This represents a continuous process, rather than a sudden transformation, involving taking action towards personally meaningful power-oriented goals – such as engaging in a community initiative, withdrawing from external influences, or asserting boundaries in intimate relationships – followed by reflection on the impacts of these actions and, crucially, on the role of socio-political-economic contexts and related power dynamics, subsequently enabling individuals to make more informed and strategic choices towards more refined goals. Young people interviewed for this study found strength in asserting a more overt sense of agency, making conscious choices to set boundaries or break repetitive patterns, reflecting a more empowered stance. For the YPRAs, this process was likely influenced by their engagement with the Siyaphambili Youth project, with each phase, interaction or even the recruitment process itself providing opportunities for action and new experiences prompting reflection. Many spoke of the importance of support systems, including chosen family, as well as wider community networks, and how this allowed them to access feelings of acceptance and care. For some YPRAs, access to resources – in the form of knowledge, awareness, social networks, sport, spirituality or religious outlets – enabled them to move out of survival mode, into sense-making or, for a minority, meaning-making. Empowerment must extend beyond the individual to recognise and amplify the collective resilience, survival and advocacy efforts already enacted by communities. This requires using participatory approaches to co-produce interventions, honouring the expertise and agency of communities. By working in true partnership with

communities, initiatives are not only more contextually relevant and sustainable, but the process of creation and implementation itself can be empowering, fostering a greater sense of agency and autonomy as a collective.

## 4.2. Reflections

Our team of researchers and providers brought together a range of positioned perspectives across lived and learned expertise, situated within and outside of the context. Bringing these multiple lenses together, combined with purposefully using a Black feminist lens allowed us to consider various dimensions of experience in our analysis. However, we are mindful of several limitations to our interpretive process; Firstly, given that the interviews were translated and analysed in English, it is likely that some nuance of language as well as culture- and language-bound idioms may have also been lost in translation. Secondly, the YPRAs were not involved in the interpretation of the data and our interpretations are inevitably shaped by our own cultural, professional, and geographic positions, as such, how we have understood and attempted to bridge the language gap to enable broader understanding may not necessarily reflect the YPRAs understanding of themselves [76]. Thirdly, local discourses and young people's use of language can be influenced by interactions with researchers, service providers and exposure to narratives shaped by international development organisations which may blur the distinction between grounded local experiences and imported concepts or ways of communicating. Additionally, we acknowledge an inherent tension in our approach: while intending to honour young peoples' worldviews, the language we use to bridge academic and community perspectives may not always resonate with those same young people. In this paper we have endeavoured to expand understandings of survival and coping contextualised within socio-political-historical realities, however, we recognise this work is ongoing, as we are confronting deeply engrained narratives in mental health discourse that predominantly emphasise the individual, as such those familiar with individualistic framings may initially struggle to engage with our approach.

While this analysis used narratives that were not mental health specific, the open-ended nature of the exercises allowed YPRAs agency in choosing what they shared, facilitating greater insight into the spectrum of experiences, potentially bypassing internalised narratives around 'what is mental health' or judgement around 'good' or 'bad' ways to cope. As a contextually grounded analyses, the emotional experiences and spectrum of coping strategies employed may not be directly reflective of other communities, however, the implications may still be beneficial to consider in other low-resource, high adversity settings. Our stance of non-judgment and contextualisation, shifting away from rigid ways of categorisation may provide further insight into the ways individuals and communities navigate survival and bolster learnings around implications for interventions in other contexts. Future research could benefit from more direct involvement of local community members in data interpretation and a deeper exploration of how different knowledge systems engage with and potentially misinterpret each other in mental health research. Additionally, while our initial analyses began to explore gender differences, we found that the themes were not distinctly gendered, with significant overlap across genders indicating shared experiences and coping strategies, and our sample size did not allow for a more comprehensive gender analysis. Given the prevalence of gender-based violence and entrenched narratives around gender roles, it would be beneficial for future research to explore this more extensively.

## 5. Conclusion

This project explored young adults' complex experiences of emotional distress, trauma and coping strategies to survive in a context where safety and structured support systems are largely absent. By taking a purposeful stance of non-judgement, we shift away from binaries around risk and resilience, holding space for the complexity of what it means to cope in the face of ongoing adversity, and provide insight into the range of practices that are employed by young adults to survive, adapt and for some, begin to transition beyond survival mode, simultaneously. Our findings reflect the complex ways young people balance survival and emotional endurance in situations of continuous trauma and underscore the importance of recognising the adaptive function of coping mechanisms often deemed harmful. By contributing to improved

understandings of lived experiences, the results add to debates around the appropriateness of applying Western-bound mental health interventions to non-Western, resource-limited settings. We advocate for interventions that integrate trauma-informed, community-based principles that empower individuals and communities to navigate ongoing trauma and foster sustainable, long-term change.

## Supporting information

**S1 Table. Codes & themes.**
(XLSX)

**S1 Checklist. *Plos* inclusivity in global research.**
(PDF)

## Author contributions

**Conceptualization:** Sorcha Ní Chobhthaigh, Rochelle A. Burgess.

**Data curation:** Smanga Mkhwanazi, Laura Washington.

**Formal analysis:** Sorcha Ní Chobhthaigh.

**Supervision:** Rochelle A. Burgess.

**Validation:** Smanga Mkhwanazi, Samantha Willan, Jenevieve Mannell, Laura Washington, Andrew Gibbs, Rochelle A. Burgess.

**Writing – original draft:** Sorcha Ní Chobhthaigh.

**Writing – review & editing:** Sorcha Ní Chobhthaigh, Smanga Mkhwanazi, Samantha Willan, Jenevieve Mannell, Laura Washington, Andrew Gibbs, Rochelle A. Burgess.

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
