## [Decision Letter · Decision Letter 0]

PMEN-D-25-00002

“The world is not a safe place”: Representations of Emotional Distress, Coping, and Survival among Young Adults in South Africa

PLOS Mental Health

Dear Dr. Ní Chobhthaigh,

Thank you for submitting your manuscript to PLOS Mental Health and I am very sorry for the severe delay in reaching a decision. After careful consideration of the reviewer reports, which we have now received, we feel that your work has merit but does not yet fully meet PLOS Mental Health’s publication criteria as it currently stands. Therefore, we invite you to submit a revised version of the manuscript that addresses the points raised during the review process.

Please address all of the comments raised by the reviewers, which you can find below and in the document attached. 

We look forward to receiving your revised manuscript.

Kind regards,

Karli Montague-Cardoso

Executive Editor

PLOS Mental Health

Journal Requirements:

1. Please include a complete copy of PLOS’ questionnaire on inclusivity in global research in your revised manuscript. Our policy for research in this area aims to improve transparency in the reporting of research performed outside of researchers’ own country or community. The policy applies to researchers who have travelled to a different country to conduct research, research with Indigenous populations or their lands, and research on cultural artefacts. The questionnaire can also be requested at the journal’s discretion for any other submissions, even if these conditions are not met.  Please find more information on the policy and a link to download a blank copy of the questionnaire here: https://journals.plos.org/plosmentalhealth/s/best-practices-in-research-reporting. Please upload a completed version of your questionnaire as Supporting Information when you resubmit your manuscript.

Additional Editor Comments (if provided):

Reviewers' comments:

Reviewer's Responses to Questions

**Comments to the Author**

1. Does this manuscript meet PLOS Mental Health’s publication criteria?

Reviewer #1: Yes

Reviewer #2: Yes

2. Has the statistical analysis been performed appropriately and rigorously?

Reviewer #1: N/A

Reviewer #2: N/A

3. Have the authors made all data underlying the findings in their manuscript fully available (please refer to the Data Availability Statement at the start of the manuscript PDF file)?

Reviewer #1: No

Reviewer #2: No

4. Is the manuscript presented in an intelligible fashion and written in standard English?

Reviewer #1: Yes

Reviewer #2: Yes

Reviewer #1: As outlined by the authors it would not be suitable to make the data for this study openly accessible. This article is well written, and the research is ethically and methodologically rigorous. I have attached my detailed feedback on the article in the attached document.

Reviewer #2: A well-written manuscript that sensitively addresses the important topic of trauma among young people in socially disadvantaged settings and provides some really interesting reflections on the limitations of classic interpretations of trauma and PTSD. A few shortcomings in the interpretation of the social context (mainly around description of contemporary South African society, and the glaring absence of a gendered lens) and some minor lack of clarity in presentation of the findings need to be addressed before publication, however.

Introduction

A more comprehensive definition of ‘trauma’ is needed, with citations. The one currently used (pg 4) is circular and doesn’t tell us much (trauma is the internal response to a traumatic event).

The absence of mention of gender-based violence in this setting is problematic; “violence” is discussed as if it is almost random but there is a strong gendered element to the forms that violence takes here and who the perpetrators and victims are.

The claim that “apartheid systems of governance and management” ended in 1991 needs qualification. In reality, the dismantling of apartheid legislation took many years following the unbanning of the ANC and the inception of democratic governance in 1994. But to think of “apartheid” only in its narrow sense of racist laws is also problematic because it shaped every aspect of South African society, and even after 1994, this continues to be the case – the most obvious example being in the fact that residential zones set out along racist lines pre-1994 continue to be organised in this way, even if not legislated as such. While the broader argument in the article about the social/structural causes of psychological trauma and the ongoing nature of trauma suggest that the authors understand this point (about apartheid not ending neatly in 1991), it feels disingenuous to make a statement like this and leave it at that.

Methods

Page 9, line 197 – “the data used in this analysis” is ambiguous. Do you mean the data used in the analysis for the Siyaphambili Youth project? Or the data used in the analysis for this article?

The repetition of certain descriptions of methods used makes for very confusing reading. For example, the in-depth interviews are mentioned in section 2.2 and again in 2.3 (they fit better in 2.3 alone).

Page 12, lines 233 to 235 – in this description of grouping into themes etc. maybe mention that these are provided as supplementary material?

Sub-section on Reflexivity – to be consistent, please provide race and gender details for authors SNC and JM.

Findings

Here and there, some additional context would help the reader to interpret the participant quotes. For example, page 14, line 287 – “she kept quiet because there is nothing…” - tell us more about who “she” is. In the quote from lines 290 to 294, who is “they” and “she”?

It’s not entirely clear how the quote on page 15, lines 310-313 illustrate “anticipated social judgment or rejection, and a sense of feeling weighed down by pressures and expectations”. Perhaps provide a second quote to illustrate that point, since the one provided (lines 310-313) only really reflects the lack of physical safety for women.

Some of the description on page 17 for the global theme on being mugged comes across as a bit euphemistic – by the phrase “by any means necessary”, are you essentially referring to involvement in crime/theft? Much of the text for this theme is not really convincing. The 2 quotes at the top of page 18 seem to reflect a deliberate choice to be passive and avoid intervening in situations where they witness violence, which suggest an underlying fear of getting involved (as this might be dangerous?) - which doesn’t really resonate with the title of this global theme of “finding ways to obtain resources by any means necessary”, or of being “trapped in the cycle together”.

It's not clear what is meant by “societal resistance” on page 23, line 480. Resistance to what? Who is resisting?

Discussion

“These experiences echo Post-Traumatic Growth” – what is Post-Traumatic Growth? In what ways do these experiences echo it? The claim is made without explanation or backing up.

Page 23, line 498 – I take your point that there should be focus on navigating the present, but perhaps “processing the past” should not be jettisoned entirely? I would add “only” just before “processing…”. A similar point could be made of “healing the wounds”, top of page 24.

Page 24, line 520 – after “compassionate approach”, add the phrase “in the context of ongoing trauma” – as a way to link your recommendation re harm reduction with your broader argument around trauma without end.

Your points towards the bottom of page 26 about how the young people could adopt “a more empowered stance” beg the question of how they were able to do this amidst what you have described in the article as truly challenging circumstances. What was the “catalyst” that prodded them towards this empowerment? The sentences that follow this raise the possibility that “support systems” and “wider community networks” may have allowed them to “access feelings of acceptance and care”. Is this what is required, then, for people to rise above the oppressive and otherwise disempowering conditions of their social existence in these settings? We don’t really get a sense of how the authors understand “empowerment” and what is needed for it to unfold. We are told what “empowerment” IS – by reference to the text linked with citation #64 – but not how one gets there. It is also not clear whether the intervention (or even just the process of taking part in the study) actually played a role in making this “empowerment” more possible among these young people.

A key gap in the Discussion and in the article overall is the lack of attention paid to gender. There are many references to gender-based violence, and we know that the setting is one where prevalence of GBV is high and culturally, there are some very disempowering ideas about femininity in traditional (and even not so traditional) gender roles. Yet the authors are silent on this, which seems odd. In the Findings, the young people are discussed as if they are a homogenous group, but I would have liked to have seen a disaggregation of experiences at least by gender. Surely the experiences (and coping strategies) for young women are different to those of their male peers?

There is mention of “our framework” in the Limitations section, which made me go back and re-read the Discussion to search for this framework. Could the authors consider developing a diagramme to represent their framework?

Limitations

Page 27, line 578 – “Although discussed within the team” – is this the research team or the team of YRPAs?

Across the manuscript, there are a number of typos and simple punctuation or grammatical errors that need fixing:

Pg 4, line 84 – “In casesopen-ended trauma”

Pg 5, line 92 – “sexist” should be “exist”

Pg 5, line 93 – “it’s” should not have an apostrophe (only the contraction of “it is” has one, not the possessive “its”)

Pg 6, line 113 – “synergistc”

Pg 6, line 125 – remove comma after “context”

Pg 7, line 131 – replace “be” with “are not”

Pg 7, line 132 – “evokes” should be “evoke”

Pg 7, line 151 – remove comma after “distress”

Pg 8, line 155 – “counter hegemonic” needs a hyphen and remove the comma after “hegemonic”

Pg 8, line 170 – add a comma after “literature”

Pg 8, line 171 – replace “their” with “one’s”

Pg 8, line 174 – add the word “that” after “argues”

Pg 10, line 198 – replace the word “completed” with “collected”

Page 10, line 199 – Full-stop after the word “lives”. New sentence beginning “This included…”

Page 10, line 215 – the word “of” is missing between “experiences” and “overlapping”

Page 10, line 218 – change “All YPRAs were not in education…” to “None of the YPRAs were in education…”

Table 1 – Add a footnote to explain that “Matric” is the final year of secondary education.

Page 12, lines 238-240: Suggested rewording of sentence – “These efforts seek to limit the positioning of researcher perspective as truth over and above the narratives…”

Page 15, line 303 – place a full-stop after “alive” and start a new sentence with “Maybe”.

Page 15, line 309 – add the words “and a” between “rejection” and “sense”

Page 15, line 315 – place a full-stop after “life”, and start the next sentence with “This was illustrated by the fourth global theme…”

Page 15, line 319 – add the word “believing” – “keeping secrets and believing that it is safer…”

Page 16, line 334 – add the word “and” after “future” and before “appreciation”

Page 17, line 344 – amend to “…I was asking [the] ancestors [for the things that] I want…”

Page 17, line 355 – add a comma and the word “as” between “needs” and “captured”.

Page 17, lines 358-360 – the sentence beginning “This cycle of shared experience…” doesn’t make sense, grammatically (especially the part at the end – “also alludes to unspoken agreements”.

Some of the participant quotes need simple punctuation marks (commas and full-stops) to make the meaning clear. For example, page 16, line 323 – “…I can tell that, no, if they were to know, I’d end up…” In the quote below this (lines 326-329), the use of the phrase “jumped off” twice in two different contexts/meanings (?) is a bit confusing.

Insert quotation marks around “Hello, my girlfriend” and other instances of direct speech in the quote on page 18, lines 380 etc.

Page 19, line 405 – Remove comma after “from”

Page 20, line 413 – add “[drugs]” after “high on them”

Page 20, lines 424-426 – simplify this sentence, which is quite hard to follow

Page 20, line 430 – the phrase “all stakes rely with the person who is abused…” doesn’t make sense

Page 21, line 450 – the word “to” is missing between “individuals” and “process”. This sentence is also very long and would be easier to understand if broken up into 2 sentences.

Page 23, line 486 – Suggested rewording: “…associated with PTSD diagnoses. Instead, they remain in a continuous cycle…”

Page 23, line 487 – add the word “as” between “successes,” and “young”

Page 23, line 488 – add the word “being” before “heard”

Page 23, line 493 – replace “on who” with “to whom”. Add “writers within” before “cultural and critical”

Page 24, line 506 – add the word “which” between “use” and “while”.

Page 24, line 507 – remove the comma after “distress”

Page 24, line 510-511 – put the phrase “can lead to additional harm” right at the end of the sentence – i.e., “…to stop using these strategies, without ensuring that….. can lead to additional harm.”

Page 24, line 517 – remove the comma after “behaviours”

Page 25, line 525 – add “and” after “housing” and before “income security”

Page 25, line 540 – add comma after “settings”

Page 26, line 553 – end the sentence after “realities”. New sentence beginning with “As such,…”

Page 26, line 560 – add the phrase “interviewed for this study” after “Young people”

Page 27, line 576 – add the word “that” after the word “Given”

Page 27, line 585 – add comma after “low-resource”

**Do you want your identity to be public for this peer review?** For information about this choice, including consent withdrawal, please see our Privacy Policy

Reviewer #1: No

Reviewer #2: No

---

## [Decision Letter · Decision Letter 1]

“The world is not a safe place”: Representations of Emotional Distress, Coping, and Survival among Young Adults in South Africa

PMEN-D-25-00002R1

Dear Ms Ní Chobhthaigh,

We are pleased to inform you that your manuscript '“The world is not a safe place”: Representations of Emotional Distress, Coping, and Survival among Young Adults in South Africa' has been provisionally accepted for publication in PLOS Mental Health.

Best regards,

Karli Montague-Cardoso

Staff Editor

PLOS Mental Health

Reviewer Comments (if any, and for reference):

Reviewer's Responses to Questions

**Comments to the Author**

Reviewer #1: All comments have been addressed

publication criteria?

Reviewer #1: Yes

3. Has the statistical analysis been performed appropriately and rigorously?

Reviewer #1: N/A

4. Have the authors made all data underlying the findings in their manuscript fully available (please refer to the Data Availability Statement at the start of the manuscript PDF file)?

Reviewer #1: No

5. Is the manuscript presented in an intelligible fashion and written in standard English?

Reviewer #1: Yes

Reviewer #1: (No Response)

**Do you want your identity to be public for this peer review?** For information about this choice, including consent withdrawal, please see our Privacy Policy

Reviewer #1: **Yes: ** Clare Coultas
